# The Relationship of Age and BMI with Physical Fitness in Futsal Players

**DOI:** 10.3390/sports7040087

**Published:** 2019-04-15

**Authors:** Pantelis T. Nikolaidis, Hamdi Chtourou, Gema Torres-Luque, Thomas Rosemann, Beat Knechtle

**Affiliations:** 1Exercise Physiology Laboratory, 18450 Nikaia, Greece; pademil@hotmail.com; 2Institut Supérieur du Sport et de l’éducation physique de Sfax, Université de Sfax, Sfax 3000, Tunisie; h_chtourou@yahoo.fr; 3Activité Physique: Sport et Santé, UR18JS01, Observatoire National du Sport, Tunis 2020, Tunisie; 4Faculty of Humanities and Science Education, University of Jaen, 23071 Jaén, Spain; gtluque@ujaen.es; 5Institute of Primary Care, University of Zurich, 8091 Zurich, Switzerland; thomas.rosemann@usz.ch; 6Medbase St. Gallen Am Vadianplatz, 9001 St. Gallen, Switzerland

**Keywords:** body fat, football, jump, overweight, sport-related physical fitness, team sport

## Abstract

The aim of this study was to examine the relationship of age and body mass status with field and laboratory measures of physical fitness in futsal players. Futsal players (*n* = 65, age 12.9 ± 2.8 years), who were classified into U11 (*n* = 28, 9–11 years), U13 (*n* = 21, 11–13 years), and adults (*n* = 16, >18 years), performed a physical fitness battery consisting of both laboratory and field tests. A similar prevalence of overweight (25%) was observed in all age groups (χ^2^ = 1.94, *p* = 0.380, φ = 0.17). Age groups differed for all parameters, except body fat percentage, with adult players showing higher values than the younger groups (*p* < 0.05). U13 was heavier, taller, and had larger fat-free mass than U11 (*p* < 0.05). Adult players had superior values than their younger counterparts for all physical fitness parameters (*p* < 0.05). Body mass index (BMI) correlated inversely with aerobic capacity (U13), jumping ability, relative isometric muscle strength, and relative mean power in the Wingate anaerobic test (WAnT) (U11) (*p* < 0.05). Also, it correlated directly with absolute isometric muscle strength (U11) and peak power, mean power (all groups), and fatigue index (U11, U13) in WAnT (*p* < 0.05). Considering the results of this study, it was concluded that the prevalence of overweight in futsal players should be an important concern for practitioners working in this team sport. Optimizing BMI should be considered as a training and nutrition goal in order to improve sport performance.

## 1. Introduction

Futsal is an indoors team sport relying on high-intensity intermittent activities with increased demands for aerobic capacity, sprint ability, and leg muscle power [1,2]. With regards to anthropometric characteristics, futsal players were characterized by values of height, body mass, body mass index (BMI), and body fat percentage (BF) similar to those of soccer players [3,4,5]. Recent studies have suggested anthropometric measurements might be related to physical fitness components in team sports [6,7,8]. For example, high body mass and BF measurements were related to poor muscle power in soccer [6], basketball [7], and handball [8] players. 

In addition, a sport-specific characteristic of team sport players [9] and one of the training and nutrition concerns in futsal is the optimization of body mass. An assessment method to monitor body mass status was body mass index (BMI), which has been used to categorize humans as underweight, normal weight, overweight, and obese for health purposes. BMI was included in several physical fitness batteries administered to futsal players [3,4,10,11]. Nevertheless, the relationship of BMI with other physical fitness components has not been examined in futsal so far. Moreover, it has been observed that the prevalence of overweight classification might vary by age (i.e., lower prevalence in the older age groups) in team sports, such as soccer [12]. Accordingly, it was shown that the magnitude of the relationship between BMI and physical fitness was smaller in the older age groups of basketball players [7].

Although the abovementioned research enhanced our understanding of the relationship between anthropometry and physical fitness in a wide range of team sports [6,7,8], this topic has not been studied in futsal so far. Considering the popularity of futsal, this information would have practical applications for practitioners working with futsal players (e.g., coaches and fitness trainers). It should be highlighted that physical fitness components such as aerobic capacity and sprint have been shown to discriminate futsal players of different competitive levels [13]. 

Therefore, the aim of this study was to investigate the relationship of BMI with laboratory and field measures of physical fitness in different age groups of futsal players. The research hypothesis was that high scores of BMI would be associated with lower levels of physical fitness and this association would be smaller in adult futsal players.

## 2. Materials and Methods

A cross-sectional research design was used to study the association of BMI with physical fitness in futsal players of three age groups. The outcomes of fitness tests were dependent variables, whereas BMI status and age group were independent variables. All study procedures were carried out in agreement with the Declaration of Helsinki and the local Institutional Review Board approved them (EPL 2013-1). Informed written consent from participants (and parents or guardians in the case of children) was provided. 

All participants (*n* = 65) were futsal players of AEK Athens and were members of the under 11 years (U11, age 8.9–10.9 years; *n* = 28), under 13 years (U13, 11.0–12.9 years; *n* = 21), and adult team (18.0–36.3 years; *n* = 16). The adult team competed in the top national league, whereas the U11 and U13 teams competed in regional (Attica) leagues since no national league existed for these age groups. Exclusion criteria included the existence of any chronic disease or orthopedic condition inhibiting the participation in the training program, matches of their team, and the realization of the tests. Participants were classified as normal-weight and overweight, and a series of physical fitness tests was administered in the laboratory (Day 1) and in the field (Day 2). Their sport experience was at least three years and they routinely competed in one match during weekend. U11, U13, and adults participated, respectively, in three, four, and five training sessions weekly. The testing sessions (Day 1 and Day 2) were carried out in the beginning of the competitive period of season 2013–2014 (September 2013).

It should be noted that participants were familiar with the administered tests, since they had routinely performed such tests in their training programs. A 24-h rest was provided between Day 1 and Day 2. In Day 1, participants were examined for anthropometric characteristics, body composition, jumping ability, muscle strength, and anaerobic power in the laboratory; whereas Day 2 included sprint and aerobic capacity tests in the field. The order of the tests for all participants was the same to elicit similar fatigue.

To assess anthropometry and body composition, a body mass scale (HD-351 Tanita, Illinois, USA), a portable stadiometer (SECA, Leicester, UK), and a caliper (Harpenden, West Sussex, UK) measured body mass (to the nearest 0.1 kg), height in the Frankfurt plane (1 mm), and skinfolds’ thickness (0.2 mm), respectively. BMI was calculated by the formula ‘body mass (kg) to height squared (m^2^)’. Parizkova’s formula (BF = −41.32 + 12.59 × log_e_x, where x the sum of 10 skinfolds) using 10 skinfolds (cheek, wattle, chest I, triceps, subscapular, abdominal, chest II, suprailiac, thigh, and calf) was utilized to estimate BF [14]. Fat mass (FM) was calculated as the product of body mass and BF, and fat free mass (FFM) as the difference ‘body mass - FM’. A table of decimals of year was used to estimate chronological age for each participant [15].

Low back and hamstring flexibility were evaluated using the sit-and-reach test (SAR) [16], which was carried out on a box with 15 cm advantage. Participants performed SAR twice with a 1 min break between trials, and the highest score was recorded. Although flexibility has not been considered as a sport-related physical fitness component, it has been widely used previously in futsal [17,18]. In addition, the handgrip muscle strength test (HG) was performed for each hand using an isometric dynamometer (Takei, Tokyo, Japan), and the sum of right and left HG was calculated in absolute (kg) and relative values (kg·kg^−1^ of body mass). Participants were also tested in the Abalakov jump (AJ), i.e., countermovement jump with arm swinging [19]. Their initial position was standing with both feet together and they were requested to perform maximal jump using a fast countermovement. Participants were asked to self-select the depth of the countermovement and were advised to land close to the point of take-off. The Opto-jump (Microgate Engineering, Bolzano, Italy) was used to calculate height of jump considering the flight time [20]. Vertical jumping tests have been previously used in futsal players [13,21].

A photocell system (Brower Timing Systems, Utah, USA) timed 20 m sprint. The sprint ability has been examined often in futsal players previously [22,23]. Three pairs of photocells were placed at 0, 10, and 20 m, respectively, allowing monitoring performance for split 0–10 m, split 10–20 m, and 20 m sprint. Participants performed the test starting from a standing position at 0.5 m behind the first pair of photocells. Two trials were performed for jump and sprint tests, and the best one was reported. The 20 m shuttle run test was administered to examine endurance performance [24], in which participants performed an incremental running test between two lines 20 m apart. The speed started at 8.5 km·h^−1^ and augmented by 0.5 km·h^−1^ every minute until exhaustion. Heart rate was monitored during the endurance test using Team2 Pro (Polar Electro Oy, Kempele, Finland) and the maximal heart rate (HR_max_) was recorded. 

The statistical software IBM SPSS v.20.0 (SPSS, Chicago, USA) was used for statistical analyses. Descriptive statistics (mean and standard deviations of the mean, *SD*) were used. Participants were grouped into normal, overweight, or obese according to international age-specific cut-off points of BMI [25] corresponding to adult values proposed by the World Health Organization (normal (≤25 kg·m^−2^), overweight (25–30 kg·m^−2^), or obese (>30 kg·m^−2^) [26]. Considering the small sample size of obese, these participants were added in the overweight group. The association of overweight with age groups was investigated by chi-square. Normality was verified using Kolmogorov–Smirnov test for *n* > 50. Differences among age groups (U11, U13, and adults) were examined by a one-way analysis of variance (ANOVA) and Bonferroni post-hoc test. Eta square—classified as small (0.01 < η^2^ ≤ 0.06), medium (0.06 < η^2^ ≤ 0.14), and large (η^2^ > 0.14)—estimated effect sizes (ES) for differences in the ANOVA [27]. Differences in variables between normal and overweight participants for each age group were tested by independent *t*-test. The relationship of BMI with physical fitness was tested by correlation coefficient (*r*). The magnitude of *r* was classified as trivial (*r* ≤ 0.1), small (0.1 < *r* ≤ 0.3), moderate (0.3 < *r* ≤ 0.5), large (0.5 < r ≤ 0.7), very large (0.7 < *r* ≤ 0.9), nearly perfect (*r >* 0.9), and perfect (*r =* 1.0) [27]. Alpha level was set at 0.05. 

## 3. Results

The number (n and %) of normal weight and overweight futsal players in total and by age group can be seen in Table 1. No body mass status × age group association was observed (χ^2^ = 1.94, *p* = 0.380, φ = 0.17). 

### 3.1. The Role of Age Group

The differences in anthropometric characteristics and physical fitness by age group can be found in Table 2 and Table 3, respectively. In anthropometric characteristics and body composition, age groups differed for all parameters, except BF, with adult players showing higher values than the younger groups. The largest magnitude of differences was observed in FFM. In addition, U13 was heavier, taller, and had larger FFM than U11. With regards to physical fitness, adult players had superior values compared to their younger counterparts for all parameters, except FI of the WAnT, and their HR_max_ in the end of the endurance test was lower than U11 and U13. The largest magnitude of differences was shown in Ppeak and Pmean of the WAnT, and the lowest in relative handgrip strength.

### 3.2. The Role of BMI

The comparison between BMI groups showed that overweight players had more BF and FM than their normal weight counterparts, and this trend was observed for all age groups (Table 4). In the adult group, no difference was observed between BMI groups. In U11 and U13, BMI groups either did not differ (e.g., sprint 20 m) or normal weight outscored overweight players (e.g., Abalakov jump in U11 and aerobic capacity in U13) (Table 5). Figure 1 presents the relationship of BMI with physical fitness by age group. BMI correlated inversely aerobic capacity (U13), jumping ability, relative isometric muscle strength, and relative Pmean in WAnT (U11). Also, it correlated directly with absolute HG muscle strength (U11), Ppeak, Pmean (all groups), and FI (U11, U13) in WAnT. Table 6 showed the correlation of BF with physical fitness.

## 4. Discussion

The main findings of the present study were that (a) the body mass status was not associated with age, (b) adult futsal players outscored their younger counterparts, (c) normal weight outscored overweight futsal players for jumping ability (U11) and aerobic capacity (U13), and (d) BMI correlated with muscle strength and power directly when expressed in absolute values and inversely when expressed in relative to body mass values.

The lack of association between body mass status and age (Table 1) indicated that an increased BMI would be an important concern for practitioners working with futsal players independently from age. Actually, the overall prevalence of overweight (25%) in participants was similar as that previously reported in adolescent soccer players [12]. An excess of body mass might result from an excess of caloric intake or decreased energy expenditure (physical activity) or a combination of both, and it should be highlighted that decreased physical activity might be observed even in sport populations [28]. The high prevalence of overweight in the futsal players of the present study indicated that the participation to this team sport was not enough to achieve normal weight—despite the increased energy demands of futsal [29]—and consequently, nutrition and extra-sport physical activity should be considered, too. Accordingly, it was not surprising that experimental studies of the effect of futsal intervention on weight management did not report body mass decrease [30,31]. Furthermore, it should be noted that an excess of body mass might reflect an increased FFM rather than an increased FM. Independent from body composition, an excess body mass would comprise an extra mass that should be carried during most actions of training and match, and, consequently, might be associated with increased fatigue and injury risk [11]. In addition, the results of the present study confirmed the negative role of an increased FM as suggested by the correlations of high BF with poor performance in performance-related skills such as sprinting and jumping.

With regards to age-related differences in anthropometric characteristics and physical fitness, the adult players presented the highest scores. In futsal, previous studies showed small differences in physical fitness between adult players and an adolescent age group (U17) [17], and adult players and a young adult group (U20) [13]. This age-related trend implied that differences in physical fitness were important between children and adult futsal players; but not between adults and those in their late adolescence. Similar patterns were previously observed in soccer for performance in vertical jump [32], WAnT [33], and 20 m sprint [34]. With regards to the finding that normal weight outscored overweight futsal players for jumping ability (U11) and aerobic capacity (U13), it was in agreement with the inverse correlation between BMI and muscle strength and power. This outcome confirmed previous research in the general population reporting that normal-weight children and adolescents outscored their overweight peers in the vertical jump test [35].

The findings of the present study should be generalized to other futsal groups with caution, considering the specific performance-related characteristics of participants (e.g., competition level, age of adult players) and the small sample of sub-groups. In terms of competition level, considering their physical fitness [23,36], and the league where they competed, the adult participants in the present study could be characterized as national level, but not international. The strength of the study was the use of both laboratory (WAnT) and field tests (e.g., 20 m sprint) to profile physical fitness in three age groups of futsal players. In addition, a novel aspect was that the findings could be used as reference for the evaluation of physical fitness in futsal. For instance, the WAnT was used previously in futsal, only on either a limited number of futsal players (*n* = 6) [37], mixed sample with soccer players [38], women [39], or U20 players [40]. Future studies might use more detailed methods to assess body composition (e.g., DEXA, bioimpedance analysis) in order to provide information on the role of muscle mass. 

## 5. Conclusions

Considering the findings of the present study, it was concluded that the prevalence of overweight in futsal players should be an important concern for practitioners working in this team sport. In addition, it was observed that an increased BMI was related to decreased scores in key sport-related physical fitness parameters such as sprinting, jumping, and anaerobic power, especially in young futsal players. Thus, optimizing BMI should be considered as a training and nutrition goal in order to improve sport performance.

## Figures and Tables

**Figure 1 sports-07-00087-f001:**
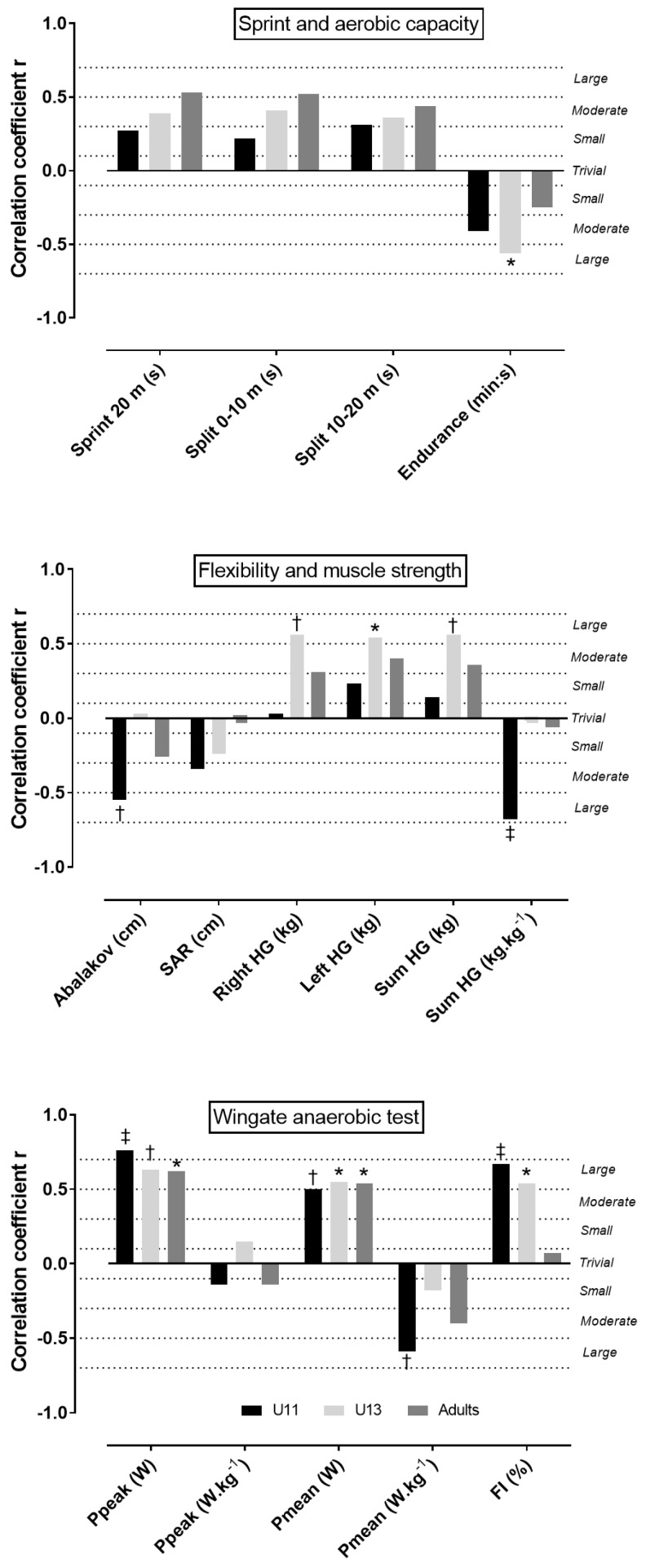
Relationship of body mass index with physical fitness. The symbols *, †, and ‡ denote significance level of *p* < 0.05, *p* < 0.01, and *p* < 0.001, respectively. SAR is sit-and-reach test, HG is handgrip muscle strength, Ppeak is peak power, Pmean is mean power, FI is fatigue index.

**Table 1 sports-07-00087-t001:** Futsal players by body mass status.

	Total (*n* = 65)		U11 (*n* = 28)		U13 (*n* = 21)		Adult (*n* = 16)	
	Normal-Weight	Over-Weight	Normal-Weight	Over-Weight	Normal-Weight	Over-Weight	Normal-Weight	Over-Weight
n	49	16	22	6	17	4	10	6
%	75	25	79	21	81	19	63	37

**Table 2 sports-07-00087-t002:** Anthropometric characteristics and body composition of futsal players by age group.

	U11 (*n* = 28)	U13 (*n* = 21)	Adults (*n* = 16)	Comparison
Age (yr)	9.9 (0.5)^¶^	11.8 (0.6)^¶^	28.9 (5.8)^§,‖^	F_2,62_ = 236.7, *p* < 0.001, η^2^ = 0.88
Body mass (kg)	36.3 (7.6)^‖,¶^	42.3 (9.4)^§,¶^	72.9 (8.3)^§,‖^	F_2,62_ = 102.4, *p* < 0.001, η^2^ = 0.77
Height (m)	1.40 (0.06)^‖,¶^	1.52 (0.11)^§,¶^	1.75 (0.06)^§,‖^	F_2,62_ = 101.3, *p* < 0.001, η^2^ = 0.77
BMI (kg·m^−2^)	18.4 (3.1)^¶^	18.1 (2.4)^¶^	24.0 (2.5)^§,‖^	F_2,62_ = 25.6, *p* < 0.001, η^2^ = 0.45
BF (%)	15.7 (5.5)	15.0 (4.0)	15.6 (3.6)	F_2,62_ = 0.1, *p* = 0.884, η^2^ < 0.01
FM (kg)	6.0 (3.3)^¶^	6.5 (2.6)^¶^	11.5 (3.5)^§,‖^	F_2,62_ = 17.1, *p* < 0.001, η^2^ = 0.36
FFM (kg)	30.3 (4.6)^‖,¶^	35.8 (7.5)^§,¶^	61.4 (6.2)^§,‖^	F_2,62_ = 140.5, *p* < 0.001, η^2^ = 0.82

Data are mean (SD). BM is body mass, BMI is body mass index, BF is body fat, FM is fat mass, FFM is fat-free mass. The symbols §,‖ and ¶ denote significant differences with U11, U13, and Adult group, respectively, according to Bonferroni test.

**Table 3 sports-07-00087-t003:** Physical fitness of futsal players by age group.

	U11 (*n* = 28)	U13 (*n* = 21)	Adults (*n* = 16)	Comparison
Sprint 20 m (s)	4.00 (0.25)^¶^	3.83 (0.22)^¶^	3.18 (0.17)^§,‖^	F_2,44_ = 42.4, *p* < 0.001, η^2^ = 0.66
Split 0–10 m (s)	2.22 (0.14)^¶^	2.16 (0.12)^¶^	1.85 (0.12)^§,‖^	F_2,44_ = 26.7, *p* < 0.001, η^2^ = 0.55
Split 10–20 m (s)	1.78 (0.12)^‖,¶^	1.67 (0.11)^§,¶^	1.32 (0.06)^§,‖^	F_2,44_ = 57.8, *p* < 0.001, η^2^ = 0.72
Endurance (min:s)	5:33 (1:19)^¶^	6:08 (1:39)^¶^	10:20 (1:20)^§,‖^	F_2,44_ = 35.6, *p* < 0.001, η^2^ = 0.62
HR_max_ (bpm)	203.9 (9.9)^¶^	201.8 (7.9)^¶^	192.2 (6.9)^§,‖^	F_2,38_ = 5.7, *p* = 0.007, η^2^ = 0.23
Abalakov (cm)	23.5 (3.8)^¶^	26.5 (4.5)^¶^	38.9 (6.1)^§,‖^	F_2,62_ = 57.3, *p* < 0.001, η^2^ = 0.65
Ppeak (W)	265 (62)^‖,¶^	354 (112)^§,¶^	777 (100)^§,‖^	F_2,62_ = 171.0, *p* < 0.001, η^2^ = 0.85
Ppeak (W·kg^−1^)	7.3 (0.9)^‖,¶^	8.3 (1.0)^§,¶^	10.7 (0.7)^§,‖^	F_2,62_ = 74.3, *p* < 0.001, η^2^ = 0.71
Pmean (W)	211 (38)^‖,¶^	289 (84)^§,¶^	585 (69)^§,‖^	F_2,62_ = 179.7, *p* < 0.001, η^2^ = 0.85
Pmean (W ·kg^−1^)	5.9 (0.9)^‖,¶^	6.8 (0.9)^§,¶^	8.0 (0.5)^§,‖^	F_2,62_ = 32.0, *p* < 0.001, η^2^ = 0.51
FI (%)	33.7 (13.0)^¶^	33.4 (9.0)^¶^	43.8 (7.3)^§,‖^	F_2,62_ = 5.5, *p* < 0.001, η^2^ = 0.15
SAR (cm)	16.0 (5.6)^¶^	13.0 (6.1)^¶^	22.5 (7.8)^§,‖^	F_2,62_ = 10.4, *p* < 0.001, η^2^ = 0.25
Right HG (kg)	18.1 (3.2)^‖,¶^	23.4 (8.0)^§,¶^	45.9 (10.0)^§,‖^	F_2,62_ = 83.3, *p* < 0.001, η^2^ = 0.73
Left HG (kg)	17.7 (3.3)^‖,¶^	22.6 (7.1)^§,¶^	44.1 (8.7)^§,‖^	F_2,62_ = 94.2, *p* < 0.001, η^2^ = 0.75
Sum HG (kg)	35.8 (6.0)^‖,¶^	46.1 (14.9)^§,¶^	90.0 (18.1)^§,‖^	F_2,62_ = 93.3, *p* < 0.001, η^2^ = 0.75
Sum HG (kg·kg^−1^)	1.02 (0.22)^¶^	1.09 (0.21)	1.24 (0.25)^§^	F_2,62_ = 5.0, *p* = 0.010, η^2^ = 0.14

Data are mean ± SD. The symbols §,‖ and ¶ denote significant differences with U11, U13, and adults, respectively, according to Bonferroni test. HR_max_ is maximal heart rate in the endurance test Ppeak is peak power, Pmean is mean power, FI is fatigue index, SAR is sit-and-reach test, HG is handgrip muscle strength.

**Table 4 sports-07-00087-t004:** Anthropometric characteristics and body composition of futsal players by body mass status.

	U11 (*n* = 28)		U13 (*n* = 21)		Adults (*n* = 16)	
	Normal-Weight (*n* = 22)	Overweight (*n* = 6)	Normal-Weight (*n* = 17)	Overweight (*n* = 4)	Normal-Weight (*n* = 10)	Overweight (*n* = 6)
Age (yr)	9.9 (0.5)	10.0 (0.6)	11.8 (0.6)	11.6 (0.9)	27.3 (5.0)	31.4 (6.6)
BM (kg)	33.4 (5.3)	46.9 (5.1) ‡	40.2 (8.4)	51.5 (8.7) *	68.8 (6.6)	79.7 (6.2) †
Height (cm)	1.39 (0.06)	1.43 (0.05)	1.52 (0.11)	1.53 (0.11)	1.75 (0.07)	1.73 (0.04)
BMI (kg·m^−2^)	17.2 (2.1)	22.8 (1.9) ‡	17.3 (1.7)	21.9 (0.5) ‡	22.4 (1.6)	26.5 (1.1) ‡
BF (%)	13.8 (4.5)	22.7 (2.3) ‡	13.9 (3.1)	19.9 (4.3) †	13.5 (2.3)	19.2 (2.2) ‡
FM (kg)	4.8 (2.3)	10.7 (2.0) ‡	5.7 (2.2)	10.0 (0.9) †	9.3 (1.6)	15.3 (2.1) ‡
FFM (kg)	28.7 (3.5)	36.1 (3.2) ‡	34.4 (6.6)	41.5 (9.5)	59.6 (6.3)	64.4 (5.3)

Data are mean ± SD. BM is body mass, BMI is body mass index, BF is body fat, FM is fat mass, FFM is fat-free mass. The symbols *, † and ‡ denote significance level of *p* < 0.05, *p* < 0.01, and *p* < 0.001, respectively, for differences between normal and overweight participants.

**Table 5 sports-07-00087-t005:** Physical fitness of futsal players by body mass status.

	U11 (*n* = 28)		U13 (*n* = 21)		Adults (*n* = 16)	
	Normal-Weight (*n* = 22)	Overweight (*n* = 6)	Normal-Weight (*n* = 17)	Overweight (*n* = 4)	Normal-Weight (*n* = 10)	Overweight (*n* = 6)
Sprint 20 m (s)	4.00 (0.25)	4.00 (0.29)	3.78 (0.18)	4.04 (0.31)	3.12 (0.14)	3.37 (0.10)
Split 0-10 m (s)	2.22 (0.14)	2.22 (0.13)	2.13 (0.10)	2.28 (0.15) *	1.82 (0.10)	1.99 (0.06)
Split 10-20 m (s)	1.78 (0.12)	1.78 (0.16)	1.65 (0.09)	1.76 (0.16)	1.30 (0.06)	1.38 (0.04)
Endurance (min:s)	5:41 (1:14)	4:40 (1:46)	6:33 (1:27)	4:12 (1:12) *	10:28 (1:30)	9:54 (0:11)
HR_max_ (bpm)	203.3 (9.9)	214.0 (0.0)	202.0 (7.8)	200.5 (12.0)	191.9 (7.0)	193.5 (9.2)
Abalakov (cm)	24.3 (3.7)	20.7 (3.1) *	27.0 (3.6)	24.5 (7.7)	40.6 (6.5)	36.1 (4.5)
Ppeak (W)	247 (54)	330 (46) †	335 (85)	435 (186)	741 (82)	837 (105)
Ppeak (W·kg^-1^)	7.4 (0.9)	7.1 (0.9)	8.3 (0.6)	8.2 (2.0)	10.8 (0.6)	10.5 (1.0)
Pmean (W)	202 (34)	246 (30) †	277 (64)	340 (145)	565 (71)	617 (59)
Pmean (W·kg^−1^)	6.1 (0.9)	5.3 (1.0)	6.9 (0.7)	6.4 (1.7)	8.2 (0.5)	7.7 (0.5)
FI (%)	31.6 (13.2)	41.7 (8.7)	32.4 (9.8)	37.9 (2.2)	43.4 (10.1)	45.1 (6.3)
SAR (cm)	17.0 (5.4)	12.7 (5.7)	14.3 (5.6)	7.4 (5.4) *	24.3 (7.3)	19.5 (8.2)
Right HG (kg)	18.1 (3.3)	18.3 (2.7)	22.2 (5.7)	28.9 (14.2)	46.8 (11.8)	44.6 (6.7)
Left HG (kg)	17.4 (3.5)	19.0 (1.9)	21.6 (5.5)	27.1 (11.9)	43.4 (10.1)	45.1 (6.3)
Sum HG (kg)	35.4 (6.5)	37.3 (3.5)	43.8 (10.9)	56.0 (26.1)	90.2 (21.7)	89.7 (11.7)
Sum HG (kg·kg^−1^)	1.07 (0.21)	0.81 (0.15) †	1.10 (0.20)	1.05 (0.28)	1.31 (0.28)	1.13 (0.17)

Data are mean ± SD. The symbols *, † and ‡ denote significance level of *p* < 0.05, *p* < 0.01, and *p* < 0.001, respectively, for differences between normal and overweight participants. HR_max_ is maximal heart rate in the endurance test Ppeak is peak power, Pmean is mean power, FI is fatigue index, SAR is sit-and-reach test, HG is handgrip muscle strength.

**Table 6 sports-07-00087-t006:** Correlation (Pearson coefficient r) of BF with physical fitness in futsal players by age group.

	U11 (*n* = 28)	U13 (*n* = 21)	Adults (*n* = 16)
Sprint 20 m (s)	0.44 *	0.52 *	0.53
Split 0–10 m (s)	0.43	0.53 *	0.47
Split 10–20 m (s)	0.43 *	0.49 *	0.52
Endurance (min:s)	−0.53 *	−0.74 †	−0.20
Abalakov (cm)	−0.69 ‡	−0.30	−0.62 *
Ppeak (W)	0.66 ‡	0.23	0.14
Ppeak (W·kg^−1^)	−0.20	−0.20	−0.43
Pmean (W)	0.38 *	0.12	0.01
Pmean (W·kg^−1^)	−0.65 ‡	−0.51 *	−0.74 †
FI (%)	0.66 ‡	0.66 †	0.01
SAR (cm)	−0.48 *	−0.24	−0.40
Right HG (kg)	0.03	0.14	−0.05
Left HG (kg)	0.19	0.23	0.10
Sum HG (kg)	0.12	0.19	0.02
Sum HG (kg.kg^−1^)	−0.65 ‡	−0.18	−0.19

The symbols *, † and ‡ denote significance level of *p* < 0.05, *p* < 0.01, and *p* < 0.001, respectively. Ppeak is peak power, Pmean is mean power, FI is fatigue index, SAR is sit-and-reach test, HG is handgrip muscle strength.

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
