# Peer review of "The Relationship of Age and BMI with Physical Fitness in Futsal Players"

_sports, 2019, doi:10.3390/sports7040087_

Round 1

Reviewer 1 Report

Lines 37-40.  Divide into two sentences as the example demonstrates. “Recent studies have suggested anthropometric measurements might be related with physical fitness components in team sports [6-8]. For example, high body mass and BF measurements were related with poor muscle power in soccer [6], basketball [7] and handball [8] players.”

Line 47. Add “classification” after overweight

Line 60. The term “weaker” is a poor descriptor (“less” would work better) as a reader could interpret this specific to strength or are the authors referring to low correlational relationship is so then use “small”?

Line 70.  Does chronic disease include diabetes, exercise induced asthma, etc, or specific to other chronic diseases. Athletes with the two diseases may be able to perform just as well on the tests but exclusion is understandable.

Line 80. Was the fitness battery randomized for test order? Please clarify and if not add the reasoning why they were not.

Methods and Results.  What was the age range of the adult team? Based on the data presented the U11 has athletes from 9 – 11 years; U13 has 12 to 13 years; and the SD is close to 6 years having the adult range from 23 to 35 years. Seems like a large gap between 13 to 23 years of age. Address in the discussion section this gap.

Discussion.

The authors address the overweight BMI but do not discuss the bodyfat % which provides a more accurate representation of muscle to fat ratio.

Body mass stay the same but there could be a shift in fat to muscle ratio. Needs to be addressed in the discussion.

Were these recreational athletes or elite level? All the reader knows is they played futsal based on age.

If recreational then how do the results compare to a higher level athlete of the same sport?

Line 201.  The authors have two adult groups listed?  U17 would be considered end of pubertal growth or early adulthood. Age greater than 18 can be considered adult unless using the Futsal sport organizational definition of adult. Please clarify.  

Author Response

Lines 37-40.  Divide into two sentences as the example demonstrates. “Recent studies have suggested anthropometric measurements might be related with physical fitness components in team sports [6-8]. For example, high body mass and BF measurements were related with poor muscle power in soccer [6], basketball [7] and handball [8] players.”

Answer: We agree with the expert reviewer and revised it as suggested.

Line 47. Add “classification” after overweight

Answer: We agree with the expert reviewer and corrected it as suggested.

Line 60. The term “weaker” is a poor descriptor (“less” would work better) as a reader could interpret this specific to strength or are the authors referring to low correlational relationship is so then use “small”?

Answer: We agree with the expert reviewer and revised it as suggested (“lower level of physical fitness and this association would be smaller”).

Line 70.  Does chronic disease include diabetes, exercise induced asthma, etc, or specific to other chronic diseases. Athletes with the two diseases may be able to perform just as well on the tests but exclusion is understandable.

Answer: We agree with the expert reviewer and clarified that - as it is mentioned in the text- “the existence of any chronic disease or orthopedic condition inhibiting the realization of the tests”. Thus, if a participant has exercise-induced asthma and participates in all training activities of his team and in the match, then he is eligible to participate in this study (“the participation in the training program and match of their team, and”).

Line 80. Was the fitness battery randomized for test order? Please clarify and if not add the reasoning why they were not.

Answer: We agree with the expert reviewer and added this information (“All tests were performed in the same order in order to elicit similar fatigue for all participants.”).

Methods and Results.  What was the age range of the adult team? Based on the data presented the U11 has athletes from 9 – 11 years; U13 has 12 to 13 years; and the SD is close to 6 years having the adult range from 23 to 35 years. Seems like a large gap between 13 to 23 years of age. Address in the discussion section this gap.

Answer: We agree with the expert reviewer and added this information in the methods (“years (U11, age 8.9-10.9 years; n=28), under 13 years (U13, 11.0-12.9 years; n=21) and adult team (18.0-36.3 years; n=16)”), and discussed this issue in the discussion section (“Caution would be needed to generalize the results of this study to other futsal groups considering the specific performance-related characteristics of participants (e.g. competition level, age of adult players).”).

Discussion.

The authors address the overweight BMI but do not discuss the bodyfat % which provides a more accurate representation of muscle to fat ratio.

Body mass stay the same but there could be a shift in fat to muscle ratio. Needs to be addressed in the discussion.

Answer: We agree with the expert reviewer and addressed this issue (“Furthermore, it should be noted that an excess of body mass might reflect an increased FFM rather than an increased FM. Independent from body composition, an excess body mass would comprise an extra mass that should be carried during most actions of training and match, and, consequently, might be associated with increased fatigue and injury risk [11]. In addition, the findings of the present study confirmed the negative role of an increased FM as suggested by the correlations of high BF with poor performance in performance-related skills such as sprinting and jumping.”).

Were these recreational athletes or elite level? All the reader knows is they played futsal based on age.

If recreational then how do the results compare to a higher level athlete of the same sport?

Answer: We agree with the expert reviewer and added information on the level of participants in the methods (“The adult team competed in the top national league, whereas the U11 and U13 teams competed in regional (Attica) leagues since no national league existed for these age groups.”) and in the discussion (“In terms of competition level considering their physical fitness [23,40] and the league where they competed, the adult participants in the present study could be characterized as national level, but not international.”).

Line 201.  The authors have two adult groups listed?  U17 would be considered end of pubertal growth or early adulthood. Age greater than 18 can be considered adult unless using the Futsal sport organizational definition of adult. Please clarify.

Answer: We agree with the expert reviewer and clarified this part (“In futsal, previous studies showed small differences in physical fitness between adult players and an adolescent age group (U17) [17], and adult players and a young adult group (U20) [13].”).

Reviewer 2 Report

Summary:

A total of 65 futsal players from three age groups (U11, U13, and adult) were recruited to perform physical fitness testing. Players were categorized using BMI as normal weight or overweight in each age group and compared.

Comments:

It is not clear what age ranges were included in each age group

p.2, lines 63-64: do you mean the outcomes of the fitness tests were dependent variables?

p.4, table 1 has been referenced previously and does not need to be included with the Results section, while Tables 2 and 3 need to be placed after they are mentioned in the text. Table 3 abbreviations need to be clarified as a footnote of the table

p.5-6, Tables 5 and 6 need abbreviations clarified in the footnote

HG is not mentioned in the Methods section and is not defined in the Results section. This point needs clarification

Heart rate (HR) is mentioned as a variable in the Methods section but is not reported in the Results, other than Table 3. Was there a difference in HR between groups and between normal and overweight during these activities? Only a general HR value is provided and does not indicate how this variable was used during all activities

Figure 1 is difficult to interpret. It would be better to have these reported in tabular form or a different graphic representation

p.8, lines 183-187: these results seem expected, in general

The Discussion is very vague and does not provide any new information

Author Response

Summary:

A total of 65 futsal players from three age groups (U11, U13, and adult) were recruited to perform physical fitness testing. Players were categorized using BMI as normal weight or overweight in each age group and compared.

Comments:

It is not clear what age ranges were included in each age group

Answer: We agree with the expert reviewer and added this information in the Methods (“(U11, age 8.9-10.9 years; n=28), under 13 years (U13, 11.0-12.9 years; n=21) and adult team (18.0-36.3 years; n=16)”).

p.2, lines 63-64: do you mean the outcomes of the fitness tests were dependent variables?

Answer: We agree with the expert reviewer and clarified it (“The outcomes of fitness tests were dependent variables,”).

p.4, table 1 has been referenced previously and does not need to be included with the Results section, while Tables 2 and 3 need to be placed after they are mentioned in the text. Table 3 abbreviations need to be clarified as a footnote of the table

Answer: We agree with the expert reviewer and revised the tables as suggested.

p.5-6, Tables 5 and 6 need abbreviations clarified in the footnote

Answer: We agree with the expert reviewer and added this information.

HG is not mentioned in the Methods section and is not defined in the Results section. This point needs clarification

Answer: We agree with the expert reviewer and revised the methods (“In addition, handgrip muscle strength test (HG) was performed for each hand using an isometric dynamometer (Takei, Tokyo, Japan), and the sum of right and left HG was calculated in absolute (kg) and relative values (kg.kg-1 of body mass).”) and results sections accordingly (l.155: “...and the lowest in relative handgrip strength”; l.174: “...HG muscle strength”).

Heart rate (HR) is mentioned as a variable in the Methods section but is not reported in the Results, other than Table 3. Was there a difference in HR between groups and between normal and overweight during these activities? Only a general HR value is provided and does not indicate how this variable was used during all activities

Answer: We agree with the expert reviewer; it is mentioned in the methods that HRmax was the maximal HR in the endurance test. It was not used in other tests. We added “..., and their HRmax in the end of the endurance test was lower than U11 and U13.” in the results, and the missing information about the comparison between normal- and overweight players in table 5 (no difference was observed).

Figure 1 is difficult to interpret. It would be better to have these reported in tabular form or a different graphic representation

Answer: We agree with the expert reviewer and added explanations below it; we believe that now it is easier to interpret it.

p.8, lines 183-187: these results seem expected, in general

Answer: We agree with the expert reviewer and developed the novelty of the findings.

The Discussion is very vague and does not provide any new information

Answer: We agree with the expert reviewer and revised it to be more focused in the novel findings and practical applications.

Reviewer 3 Report

The study is performed with a good scientific background, all method used are well known and have a good history of practical value. Maybe instead of using skinfolds using bioimpedancy would be better to assess the body fat (less human-dependant method) – yet this is not a constrain of the study. Additionally this would also give the possibility to assess not only FFM but also muscle mass – as an important physical fitness and physical performance indicator. As we know often not the BF but MM is more important in physical performance in some sports.

The study present a small sample of futsal players from one futsal club. All of them train the sport for min. 3 years which can highly influence the results. The group is small (n=65) and additionally divided in to 3 subgroups by age.  This highly decreases the quality and usefulness of the results for practical use.

Although well planned methodology, well performed statistical analysis the usability of the results is poor. It does not really give any additional new knowledge to the field of sport performance, physical fitness.

Author Response

The study is performed with a good scientific background, all method used are well known and have a good history of practical value. Maybe instead of using skinfolds using bioimpedancy would be better to assess the body fat (less human-dependant method) – yet this is not a constrain of the study. Additionally this would also give the possibility to assess not only FFM but also muscle mass – as an important physical fitness and physical performance indicator. As we know often not the BF but MM is more important in physical performance in some sports.

Answer: We agree with the expert reviewer and discussed this aspect in the discussion section (“Future studies might use more detailed methods to assess body composition (e.g. DEXA, bioimpedance analysis) in order to provide information on the role of muscle mass.) “ .

The study present a small sample of futsal players from one futsal club. All of them train the sport for min. 3 years which can highly influence the results. The group is small (n=65) and additionally divided in to 3 subgroups by age.  This highly decreases the quality and usefulness of the results for practical use.

Answer: We agree partially with the expert reviewer and discussed this aspect in the discussion section as limitation (“Caution would be needed to generalize the results of this study to other futsal groups considering the specific performance-related characteristics of participants (e.g. competition level, age of adult players) and the small sample of sub-groups.”).The sample size of sub-groups is small and added this aspect in the limitations, but the whole sample of futsal players is considered among the largest in the existing literature on futsal.

Although well planned methodology, well performed statistical analysis the usability of the results is poor. It does not really give any additional new knowledge to the field of sport performance, physical fitness.

Answer: We agree with the expert reviewer about the “well planned methodology, well performed statistical analysis” and developed the aspect of usability and novelty in the paragraph before conclusions.

Round 2

Reviewer 1 Report

The authors have done a good job addressing the areas needed to clarify their points.

Reviewer 2 Report

Overall, the authors have improved this manuscript and provided sufficient answers to my initial concerns.